# Influence of Sol–Gel State in Smectite Aqueous Dispersions on Drying Patterns of Droplets

**DOI:** 10.3390/ma17122891

**Published:** 2024-06-13

**Authors:** Hiroshi Kimura

**Affiliations:** Department of Chemistry and Biomolecular Science, Faculty of Engineering, Gifu University, 1-1 Yanagido, Gifu 501-1193, Japan; kimura.hiroshi.b1@f.gifu-u.ac.jp; Tel./Fax: +81-58-293-2622

**Keywords:** smectite clay, aqueous dispersion, drying pattern, sol–gel dispersion state, Chinese black ink, Rhodamine 6G

## Abstract

The sol–gel state of smectite clay dispersions varies with the volume fraction of clay and electrolyte concentration. In this study, it was elucidated that the drying patterns of droplets from four types of smectite clay dispersions vary according to their sol–gel states. Droplets in the sol state exhibited a ring-shaped pattern, while those in the gel state showed a bump-shaped pattern. Near the boundary between the sol and gel states, patterns featuring both ring and bump structures were observed regardless of whether the droplets were on the sol or gel side. When guest particles or molecules were introduced into the clay dispersion, they dispersed uniformly within the system, and the drying pattern depended on the sol–gel state of the droplets. These findings suggest that the presence or absence of convection within the droplets during drying governs the drying pattern.

## 1. Introduction

The drying patterns of droplets in colloidal dispersions, exemplified by the “coffee ring” phenomenon, are commonly observed in everyday scenarios. In many cases, the macroscopic “drying patterns” are evaluated based on the top view of the patterns and their cross-sectional shapes. However, despite its apparent simplicity, the mechanism underlying the formation of drying patterns is profoundly intricate. It is evident that a deeper understanding of the mechanisms governing drying patterns is essential for engineering applications. Deegan et al. [1,2] demonstrated that particles are transported radially outward within the droplet by outward flows, leading to the deposition of particles at the periphery of the droplet, thus forming circular drying patterns. In addition to the “coffee ring” pattern, numerous other deposition patterns have been reported, exhibiting regular and precise arrangements of particles in two and three dimensions [3,4,5,6,7,8,9], which are of significant interest. Furthermore, various drying patterns have been reported even without regular particle arrangements, with these patterns being influenced by factors such as particle size [10], shape [11,12,13], size distribution [14], particle concentration [15,16], electrolyte concentration [17,18], pH [19,20], droplet size [21], substrate hydrophilicity/hydrophobicity [22,23], substrate temperature [24,25,26,27], and electric fields [28].

One of the primary reasons for the emergence of various drying patterns mentioned above is the difference in convection within droplets. Significant within-droplet convections include capillary flows driven by solvent evaporation [29] and Marangoni flows originating from surface tension gradients [30,31]. Of course, the “coffee ring” is also a pattern resulting from these convections. One of the most widely targeted goals even today is the suppression of the “coffee ring” phenomenon. This is because uniform drying is desired for precise coating and film formation. Generally, promoting Marangoni flows [32] is effective in suppressing the “coffee ring”. However, conventional droplets are typically in a sol state, making control of Marangoni flows challenging. Control of the “coffee ring” through factors such as evaporation rate [24] and pH [19,20], among others, has been reported. Investigations into the influence of particle concentration on the “coffee ring” pattern have also been conducted. In one study, an increase in concentration led to an increase in the thickness of the outer ring [15] (indicating the absence of “coffee ring” suppression), while another study reported the suppression of the “coffee ring” with an increase in concentration [16]. This discrepancy suggests that considerations from the rheological perspective, such as whether the droplet is in a sol or gel state throughout the drying process, are crucial. The author has previously reported the suppression of the “coffee ring” phenomenon through the introduction of trace amounts of smectite clay [33]. This suppression heavily relies on the sol–gel state of the droplet, without the need for adjustments in factors such as evaporation rate and pH. To the best of the author’s knowledge, there are few systematic investigations into the relationship between the sol–gel state and drying patterns. This study investigates, for the first time, the drying patterns of sol–gel-state droplets for four types of clay aqueous suspensions over a wide range of clay volume fractions and NaCl concentrations. Droplets of smectite clay aqueous dispersion, being controllable in their sol–gel state, present an optimal material for such investigations.

In the aqueous dispersion of smectite clay, gelation occurs when the clay volume fraction and electrolyte concentration exceed certain thresholds. The distribution of sol–gel regions varies depending on the type of clay. The author has previously conducted dynamic viscoelastic measurements of clay aqueous dispersions [34] and constructed sol–gel dispersion state diagrams based on the ratio of storage modulus *G*′ to loss modulus *G*′′, known as tan *δ*. Saponite (Sap), hectorite (Ht), stevensite (Stv), and fluorine-modified hectorite (Ht–F) were investigated. All aqueous dispersions gelled under conditions where the clay volume fraction, *ϕ*, was approximately 0.001 or higher, and the electrolyte concentration was approximately 0.01 mol/L or higher. Under conditions of *ϕ* = 0.001, Sap remained in a sol state at electrolyte concentrations below 0.001 mol/L, while the other clays formed gels at even lower electrolyte concentrations. Comparing the breadth of gelation regions, Sap < Ht, Stv < Ht–F. All clay aqueous dispersions used in the aforementioned study were alkaline, and it was presumed that the edge of clay nanosheets carried a negative charge similar to the face. Furthermore, comparing the transparency of gels formed by these clays, Sap < Ht, Stv < Ht–F. Cryo-SEM images revealed a three-dimensional network structure. The author concluded that the skeletal diameter strongly influenced the transparency of clay aqueous dispersion gels. The results of the investigation into the sol–gel dispersion state diagrams and transparency of smectite clay aqueous dispersions suggest the presence of “aggregation propensity” in water depending on the type of clay. In this study, the author investigated the drying patterns of clay aqueous dispersion droplets, with a focus on reporting the drying pattern of Sap due to its superior visibility. The drying patterns of droplets of other clay aqueous dispersions were highly transparent and had poor visibility; however, the characteristics of the drying patterns were captured by measuring the cross-sectional profiles. Some of the objectives of this study were to examine the drying patterns of droplets of the four types of clay aqueous dispersions and investigate their relationship with the sol–gel dispersion state. Other objectives were to observe the drying patterns of droplets when other colloidal particles or molecules are mixed with clay aqueous dispersions and to elucidate the influence of clay aqueous dispersions on drying patterns by comparing them with the drying patterns of droplets of colloidal particles or molecules in aqueous dispersion alone.

## 2. Experimental

### 2.1. Sample Preparation

Four types of synthetic smectite clay minerals, fluorine-modified hectorite (Ht–F), hectorite (Ht), stevensite (Stv), and saponite (Sap), were provided by Kunimine Industries Co., Ltd. (Tokyo, Japan). According to information provided by the manufacturer, the chemical compositions of these materials were as follows:

Sap: Na_0.33_Mg_3_(Si_3.67_Al_0.33_)O_10_(OH)_2_;

Ht: Na_0.33_(Mg_2.67_Li_0.33_)Si_4_O_10_(OH)_2_;

Stv: Na_0.16_Mg_2.92_Si_4_O_10_(OH)_2_;

Ht–F: Na_0.33_(Mg_2.67_Li_0.33_)Si_4_O_10_(F, OH)_2_.

Ht–F is a variant of Ht in which approximately 50% of the OH groups at the edges of the clay nanosheets are replaced by fluorine atoms. The clay nanosheets had a single-sheet thickness of 1 nm, with Stv having a diameter ranging from 30 to 50 nm, while the others (Ht–F, Ht, Sap) exhibited diameters between 50 and 100 nm. The density of all clays was 2.5 g/cm^3^. The *ζ*-potential of all clay aqueous dispersions in diluted states was approximately –40 mV. The *ζ*-potential indicates the electric potential at the slip plane, which is situated at a specific distance from the surface of clay nanosheets with a Stokes diameter of 15.3 nm. To investigate the effect of clay size on the drying structure of droplets, coarse-grained Sap (CG–Sap) was provided from the same manufacturer as the other clays. CG–Sap is Sap subjected to hydrothermal treatment at temperatures above 350 °C, resulting in clays with diameters ranging from 100 to 200 nm.

The clay minerals were dispersed in water purified by a Milli-Q reagent-grade system (Milli-Q Advantage A10, Millipore Co., Burlington, MA, USA), and then the dispersions were deionized for over three months by mixing with ion-exchange resin (AG501-X8(D), Bio-Rad Lab., Inc., Hercules, CA, USA) and used as the stock dispersion. A 1:1 electrolyte (sodium chloride, NaCl) was employed as the additive salt. In this study, each sample was prepared using the stock solution, left for over a day, and then measured within approximately one week. Stock suspension of Chinese ink (SE-1702, Kaimei Co., Urawa, Japan) was purchased. The weight fraction of the stock suspension was *w* = 0.145 g/mL. The clay volume fractions and added salt concentrations of the aqueous clay dispersions were prepared in the ranges of *ϕ* = 0.001–0.01 and [NaCl] = 0–0.1 mol/L, respectively. The colloidal particles in the ink were highly polydisperse, and the mean diameter was 33 ± 9 nm [35], which was determined from transmission electron microscope measurements (H8100, Hitachi Ltd., Tokyo, Japan). The stock suspension was deionized using the same method as for the clay mentioned above. Specific gravity of suspended particles of Chinese ink was decided to be 1.4 from repeated measuring of the mass of dried samples of 1 mL suspension. Rhodamine 6G (R6G) was purchased as a dye molecule from FUJIFILM Wako Pure Chemicals Co. It was dissolved in ultrapure water to prepare the stock solution, resulting in a concentration of [R6G] = 0.006 mol/L.

### 2.2. Observation of Droplet Drying Patterns

The volume of droplets for all aqueous dispersions was standardized to 0.1 mL. Droplets were carefully dispensed onto a horizontal cover glass (30 mm × 30 mm, Matsunami Glass Ind., Ltd., Osaka, Japan) using a pipette (4910, Eppendorf Co., Ltd., Singapore). For some samples, disposable pipettes were used to control the volume of droplets by mass. The cover glass was utilized without additional rinsing. The drying process was monitored using a CCD microscope or a digital camera, and the thickness of the drying pattern was measured using a laser 3D profile microscope (VK-8500, Keyence Co., Osaka, Japan). All measurements were conducted at 25 °C. The humidity was maintained within the range of 35%RH to 50%RH, centered around 40%RH.

## 3. Results and Discussion

### 3.1. Dispersion State of Saponite (Sap) in Aqueous Medium

Clay nanosheets can be considered to disperse in “deionized” water either individually or in the form of small flocs. Assuming that clay nanosheets disperse individually, the Stokes diameter, *d*_S_, for discotic particles can be described by the following equation [36]:(1)$$d_{S}=\delta \sqrt{3\tan^{-1}\rho /(2\rho )}$$

Here, *δ* represents the surface diameter, and *ρ* is the value obtained by dividing *δ* by the thickness of the disc. For Sap nanosheets, with *δ* = 100 nm and *ρ* = 100, the Stokes diameter, *d*_S_, is calculated to be 15.3 nm. Assuming only 1:1-type electrolytes are present in the aqueous medium, the thickness of the electric double layer (*L*_D_) can be determined by the following equation:(2)$$L_{D}=1/\kappa =\sqrt{\frac{\varepsilon k_{B}T}{2000e^{2}nN_{A}}}$$

Here, *κ* represents the Debye–Hückel parameter, *ε* the dielectric constant of water, *k*_B_ the Boltzmann constant, *T* the absolute temperature, *e* the elementary charge, *n* the electrolyte concentration, and *N*_A_ Avogadro’s number. Assuming *n* = 1.0 × 10^–5^ mol/L, *L*_D_ is calculated as 96 nm. While a rigorous discussion of the electrostatic interactions between clays is challenging, the author assumes the presence of an electrical double layer with *L*_D_ = 96 nm surrounding spherical particles with a diameter of 15.3 nm. In order to estimate the effective salt concentration of the dispersion medium, the electrical conductivity of ultrapure water used in this study was measured, yielding a value of 7.5 × 10^–5^ S/m. Generally, the electrical conductivity of an aqueous solution is proportional to the concentration of electrolytes in the system. For monovalent ions, for example, when the electrolyte concentration is 1.0 mol/L, the electrical conductivity of the aqueous solution is 5.2 S/m. As a result, the electrolyte concentration of a solution with a conductivity of 7.5 × 10^–5^ S/m is approximately 1 × 10^–5^ mol/L. The potential energy in the DLVO theory can be expressed as follows [37,38]:(3)$$V=V_{R}+V_{A}=\frac{\pi d^{2}\varepsilon {\Psi_{0}}^{2}exp(–\frac{H}{L_{D}})}{d+H}-\frac{A_{H}d}{24H}$$

Here, the symbols represent the following parameters: *V* is total potential energy, *V*_R_ electrostatic potential energy, *V*_A_ van der Waals attractive potential energy, *d* particle diameter, *ε* the dielectric constant of water, *Ψ*_0_ surface potential of the particle, *H* inter-particle distance, and *A*_H_ Hamaker constant. *V*_R_ is the formula applicable when *d* << *L*_D_ [39].

Figure 1 illustrates the potential curve obtained. Here, *d* has been replaced by *d*_S_ and *Ψ*_0_ by the *ζ*-potential.

The Hamaker constant for clay in water is unknown. The typical Hamaker constant for general substances ranges from 1.0 × 10^–21^ to 1.0 × 10^–19^ (J); however, within this range, it is anticipated that there exists electrostatic repulsion between clay nanosheets. Here, Sap was treated as a representative example; however, since similar potential curves were observed for other clays, it can be concluded that all ‘deionized’ clay aqueous dispersions disperse in water as individual or small flocs. Gelation of clay aqueous dispersions occurs due to an increase in the effective electrolyte concentration resulting from an increase in clay volume fraction or an increase in added electrolyte concentration, leading to a reduction in electrostatic repulsion between clay particles.

### 3.2. Drying Patterns of Droplets

The author investigated the drying patterns of droplets and corresponding cross-sectional profiles of four types of smectite clay aqueous dispersions (Figure 2).

Here, Sap is presented as a representative example as it exhibited the most opaque drying film with excellent visibility. As examples, drying patterns for *ϕ* = 0.0029 and 0.0036 are shown. When the Sap dispersion was in the sol state (Figure 2a,b,g,h), a coexistence pattern of “ring-shaped” and “ring-shaped with bump” was observed from the top view. Since individual clay particles or flocs in the water are very small, resulting in a high Péclet number, they are easily transported to the outer periphery of the droplet during drying along with convection. The phenomenon of droplets from sol-state dispersions exhibiting a ring-shaped pattern during drying has been reported by numerous researchers.

During the drying of droplets, convection within the droplet accumulates Sap near the contact line where three phases (droplet, glass surface, air) meet. On the other hand, an observation of coexisting ring-shaped and bump-shaped patterns was made as the concentration of added salt, [NaCl], increased, which is believed to be due to gelation of the Sap dispersion as drying progresses. While it is difficult to distinguish between the coexisting pattern and the ring-shaped pattern from top-view observation, it becomes clearer through cross-sectional profiles. Comparing Figure 2d (ring-shaped pattern) and Figure 2e (coexisting ring-shaped and bump-shaped pattern), it was observed that, in the ring-shaped pattern, only the edge of the drying film was thicker, whereas, in the coexisting pattern, in addition to the thick edge, the central region of the film also exhibited small bump-like features. With higher salt concentrations (0.01 mol/L), the bump-shaped pattern was manifested (Figure 2c,i). In this case, the Sap dispersion was in a gel state immediately after droplet deposition. In the gel state, convection within the droplet during drying is suppressed, and moisture evaporates from the surface of the droplet. As a result, the cross-section of the drying film appears to shrink from its initial shape immediately after deposition into a bump-like shape.

Figure 3 illustrates the boundary line of the sol–gel dispersion determined by dynamic viscoelastic measurements conducted by the author previously.

The symbols for drying patterns within the figure represent the observation results of this study. When viewed as a whole, the drying patterns in the sol region appeared as ring-shaped, while those in the gel region exhibited bump-shaped patterns. Near the boundary of the sol–gel state, all observed drying patterns were of the coexisting type, featuring both ring and bump configurations. These results near the boundary line can be explained by dividing into two regions: the lower side (sol) and the upper side (gel) of the boundary line. When the clay aqueous dispersion is in the sol region, droplets initially form a ring-shaped pattern upon deposition. However, as drying progresses, the increase in clay volume fraction and the introduction of bicarbonate ions from the air cause the undried central portion of the dispersion to gel, resulting in the manifestation of the coexisting pattern. On the other hand, when the clay aqueous dispersion is in the gel region, upon droplet deposition from the syringe, the dispersion undergoes flow deformation, temporarily transitioning into a sol state but eventually re-gelling over time. As a result, droplets on the sol side near the boundary line form the same coexisting pattern as observed. The differences in the sol–gel regions according to the type of clay seemed to be reflected in the drying patterns of droplets. The relationship between the sol–gel dispersion state determined by dynamic viscoelastic measurements in previous studies and the drying patterns of droplets could be qualitatively well explained.

Drying patterns of droplets of coarse-grained Saponite (CG–Sap) aqueous dispersion were investigated (Figure 4).

Comparing the drying patterns of CG–Sap (Figure 4a–c) with those of Sap dispersion at *ϕ* = 0.0036 (Figure 2g–i), it was found that the drying patterns of CG–Sap were highly opaque and more prone to exhibiting bump-shaped patterns. In other words, when the clay size is larger, the gelation region of the clay aqueous dispersion expands. Focusing on deionized dispersions ([NaCl] = 0), ring-shaped or coexisting patterns were observed when *ϕ* was smaller (Figure 4g–i).

Drying patterns of CG–Sap droplets within the sol–gel dispersion state diagram of Sap aqueous dispersion (Figure 3a) were depicted. The gel region of CG–Sap expanded to conditions with a smaller *ϕ* compared to the gel region of Sap (Figure 5). It became evident that the distribution of sol–gel states of smectite clay aqueous dispersions depends not only on the type of clay but also on the size of the clay particles. The distribution of sol–gel states of CG–Sap resembled that of Ht–F, which has the widest gel region. The author compared the drying patterns of droplets of CG–Sap, Sap, and Stv aqueous dispersions under the same conditions (*ϕ* = 0.0036, [NaCl] = 0) (Figure 6).

The outer periphery of Stv droplets during deposition was indicated with white dashed lines as the drying film of the droplets was nearly colorless and transparent. As mentioned earlier, CG–Sap and Sap exhibit different film transparency due to their size difference. However, even when the type of clay differed, it was evident that the transparency of the drying film changed. Clays exhibit different propensities for aggregation in water depending on their type, which may also influence the transparency of the drying film. Indeed, even when compared in transparent cells (Figure 6d), the difference in transparency between Sap and Stv aqueous dispersions was apparent. The differences in film and dispersion transparency can be interpreted as differences in Mie scattering intensity due to the presence of clay.

The drying patterns of Chinese black ink and the ink mixed with Ht aqueous dispersion are shown in Figure 7.

The drying pattern of ink alone exhibited a ring-shaped pattern even with increasing salt concentration. It is known that ink remains a highly viscous sol even at a high ink weight fraction, *w*, up to 0.3 g/mL [40]. On the other hand, when mixed with Ht aqueous dispersion (Figure 7g–i), the drying pattern exhibited clear dependence on the added salt concentration. At [NaCl] = 0, a ring-shaped pattern was observed; at 0.001 mol/L, a coexisting pattern was observed; and at 0.01 mol/L, a bump-shaped pattern was observed. Within droplets in the sol state, ink particles clearly convected within the droplet alongside the clay, while, in droplets in the gel state, the movement of ink particles was restricted. Next, the drying patterns of droplets of R6G dye and R6G mixed with Sap aqueous dispersion are shown in Figure 8.

When R6G is used alone, the outer periphery of the droplet is initially circular upon deposition; however, after drying, it tends to aggregate in certain areas, losing its initial circular shape, resulting in the formation of drying patterns. This phenomenon bears a striking resemblance to the drying patterns observed in droplets of NaCl aqueous solutions. Considering the low total salt concentration (0.01 mol/L or less), it is possible that surface tension undergoes complex changes known as the Jones–Ray effect with increasing salt concentration [41,42]. On the other hand, when mixed with Sap aqueous dispersion (Figure 8g–i), a clear dependence on added salt concentration was observed. In the low-[NaCl] region, a coexisting pattern was observed, while, at [NaCl] = 0.01 mol/L, a bump-shaped pattern was observed. R6G, like particles of Chinese black ink, exhibited convection within the sol, facilitated by the clay, resulting in coexisting patterns. Conversely, in the gel phase, despite the small size of R6G molecule, it was fully affected by the suppression of convection. The fact that R6G is strongly affected by the suppression of convection within the physical gel strongly suggests that the convection of the water itself is also being suppressed within the gel.

## 4. Conclusions

Droplets of smectite clay aqueous dispersions show distinct drying patterns based on their dispersion state. In the sol state, a typical ring pattern forms, while, in the gel state, coexisting or bump patterns appear. This suggests that convection within the droplets governs the drying patterns. Radial flow in the sol state moves clay to the outer edges, forming a ring pattern. As the droplet transitions to a gel state, bump patterns emerge. The sol–gel region, influenced by clay type and electrostatic interactions, affects the dispersion state diagram and results in varying turbidity. Mixing colloidal particles or dye molecules into the gel state inhibits convection during drying. Gelation suppresses water’s convective flow, which, in turn, inhibits the flow of guest substances. Clay aqueous dispersions, with their easy gelation control, have promising engineering applications due to their ability to suppress convective flow during drying.

## Figures and Tables

**Figure 1 materials-17-02891-f001:**
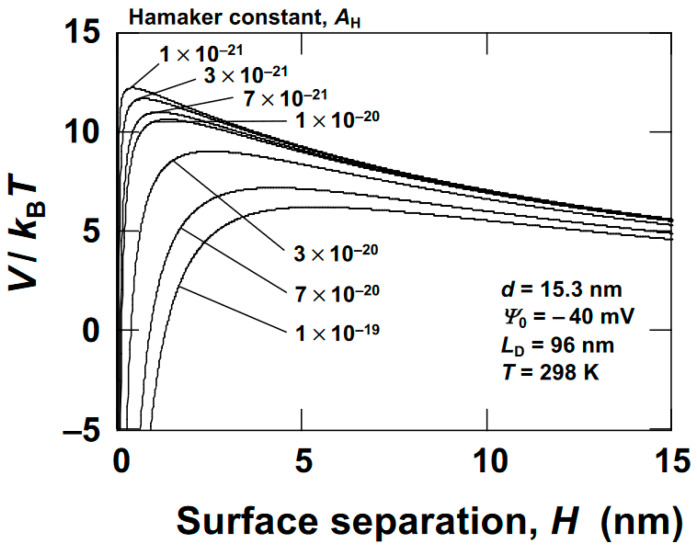
DLVO potential curves in deionized water for saponite (Sap) at 25 °C. Several curves are depicted within the range of Hamaker constants from 1 × 10^–21^ J to 1 × 10^–19^ J. *d* (*d*_S_) = 15.3 nm, *L*_D_ = 96 nm (*n* = 1.0 × 10^–5^ mol/L), *Ψ*_0_ (*ζ*-potential) = −40 mV.

**Figure 2 materials-17-02891-f002:**
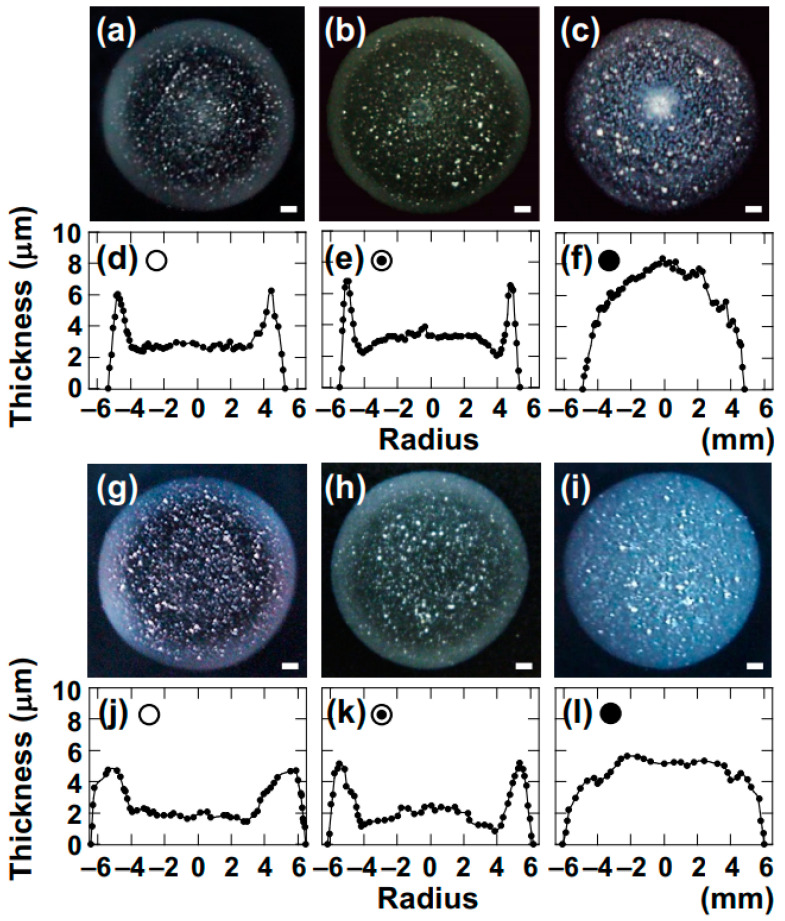
Drying patterns and cross-sectional profiles of Saponite aqueous dispersion at 25 °C. (**a**–**f**) Clay volume fraction, *ϕ* = 0.0029, (**g**–**l**) *ϕ* = 0.0036. (**a**,**d**,**g**,**j**) Salt concentration, [NaCl] = 0, (**b**,**e**,**h**,**k**) 0.001 mol/L, (**c**,**f**,**i**,**l**) 0.01 mol/L. 
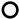
: ring pattern, 
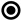
: coexistence of ring and bump patterns, and 
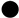
: bump pattern. Length of the bar is 1.0 mm. These symbols are based on the image observed from the top view and are shown within each cross-sectional profile figure.

**Figure 3 materials-17-02891-f003:**
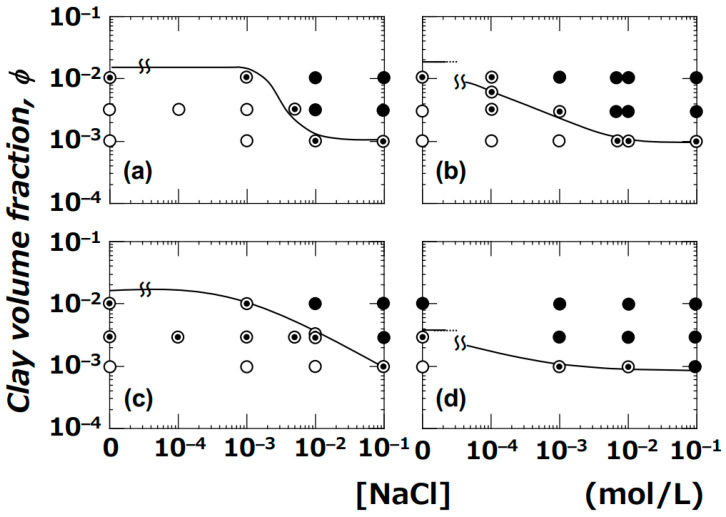
Drying patterns of four types of clay aqueous dispersions compared with the sol–gel phase diagram plotted at 25 °C. (**a**) Sap, (**b**) Ht, (**c**) Stv, (**d**) Ht–F. Solid lines represent the boundaries of the sol–gel state determined by dynamic viscoelastic measurements. 
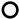
: ring pattern, 
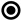
: coexistence of ring and bump patterns, and 
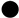
: bump pattern.

**Figure 4 materials-17-02891-f004:**
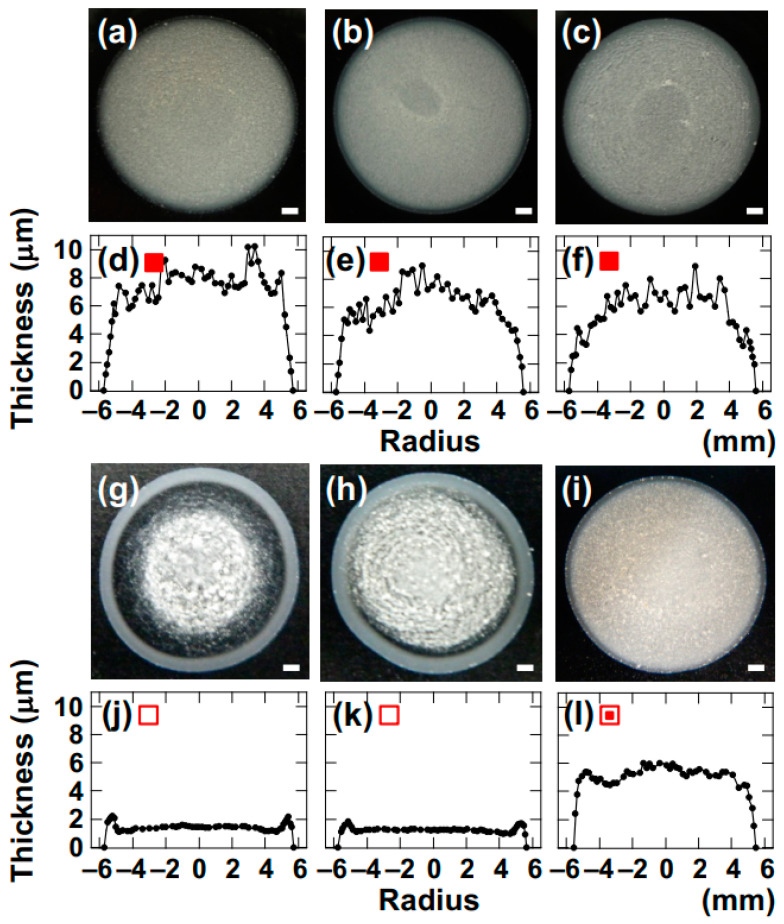
Drying patterns and cross-sectional profiles of coarse-grained saponite (CG–Sap) aqueous dispersions at 25 °C. (**a**–**f**) *ϕ* = 0.0036. (**a**,**d**) [NaCl] = 0, (**b**,**e**) 0.001 mol/L, (**c**,**f**) 0.01 mol/L. (**g**–**l**) [NaCl] = 0, (**g**,**j**) *ϕ* = 0.0005, (**h**,**k**) 0.00084, (**i**,**l**) 0.0025. The red symbols are used to distinguish them from the Sap.

**Figure 5 materials-17-02891-f005:**
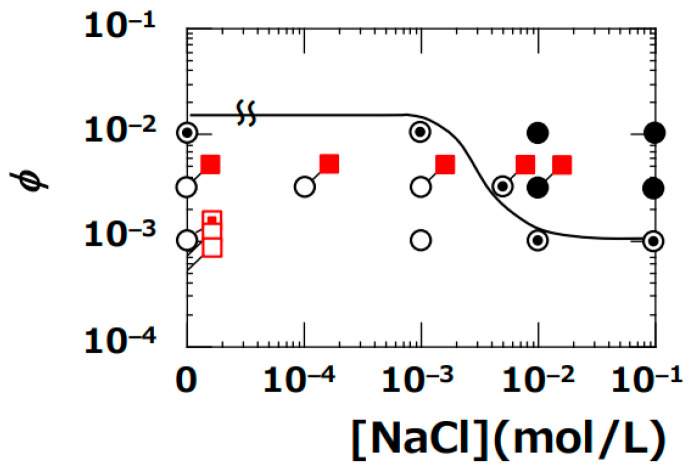
Drying patterns of CG–Sap aqueous dispersions at 25 °C. 
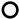
: ring pattern, 
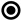
: coexistence of ring and bump patterns, 
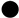
: bump pattern. Data for CG–Sap are shown on the data in Figure 3a for comparison with Sap. The red symbols indicate the drying pattern of CG–Sap.

**Figure 6 materials-17-02891-f006:**
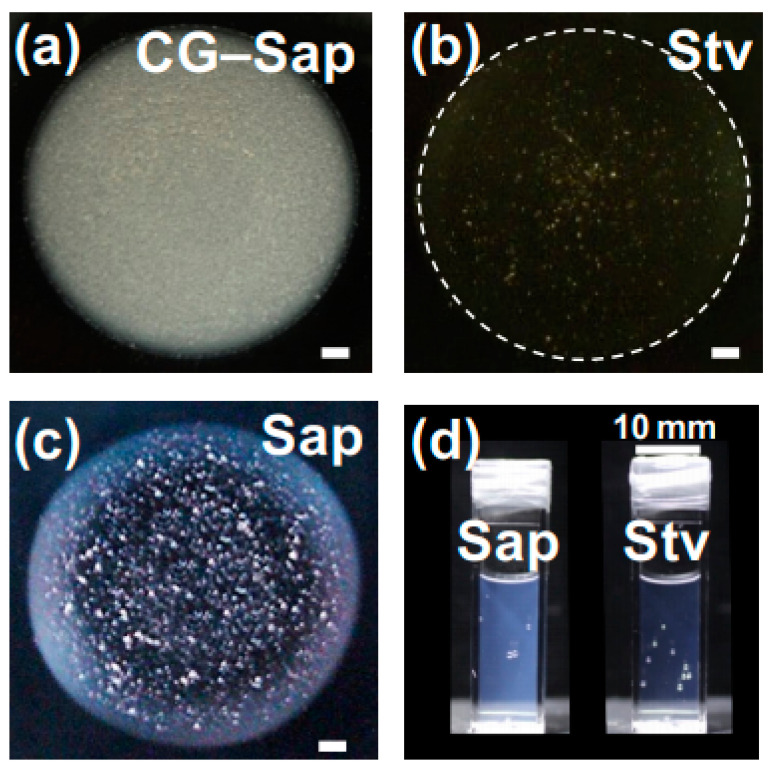
Drying patterns (**a**–**c**) and transparency comparison (**d**) of clay aqueous dispersions prepared under the same conditions at 25 °C. *ϕ* = 0.0036, [NaCl] = 0, (**a**) CG–Sap, (**b**,**d** (**right**)) Stv, (**c**,**d** (**left**)) Sap.

**Figure 7 materials-17-02891-f007:**
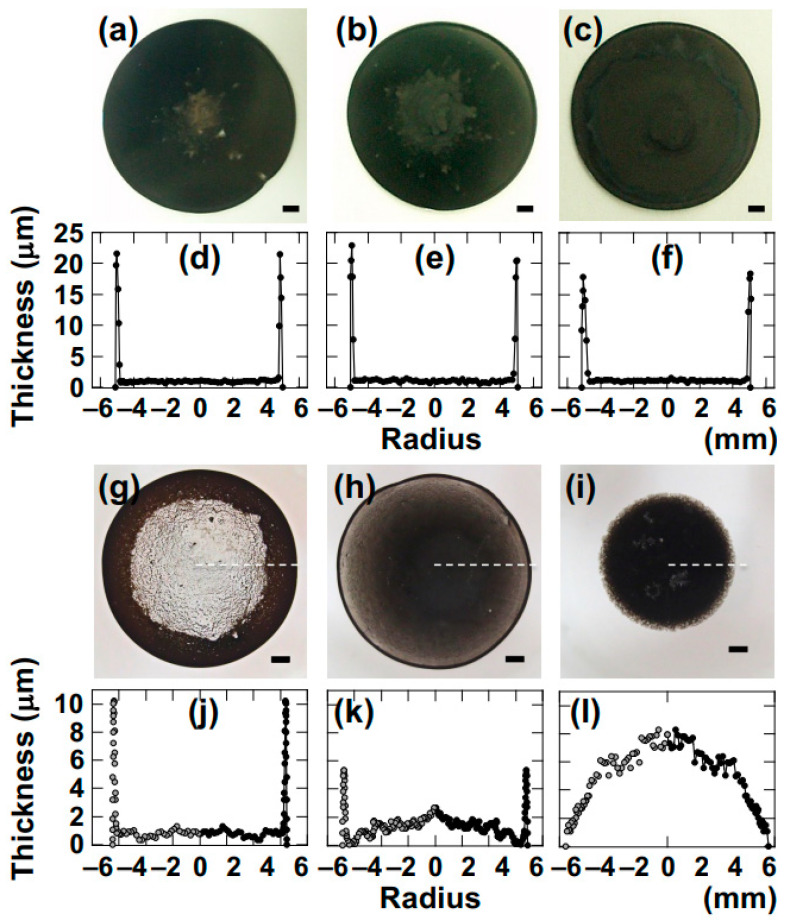
Drying patterns and cross-sectional profiles of Chinese black ink and the ink-mixed Sap aqueous dispersions at 25 °C. (**a**–**f**) *ϕ* = 0, *w* = 0.002 g/mL. (**a**,**d**) [NaCl] = 0, (**b**,**e**) 0.001 mol/L, (**c**,**f**) 0.01 mol/L. (**g**–**l**) *ϕ* = 0.0036, *w* = 0.01 g/mL. (**g**,**j**) [NaCl] = 0, (**h**,**k**) 0.001 mol/L, (**i**,**l**) 0.01 mol/L.

**Figure 8 materials-17-02891-f008:**
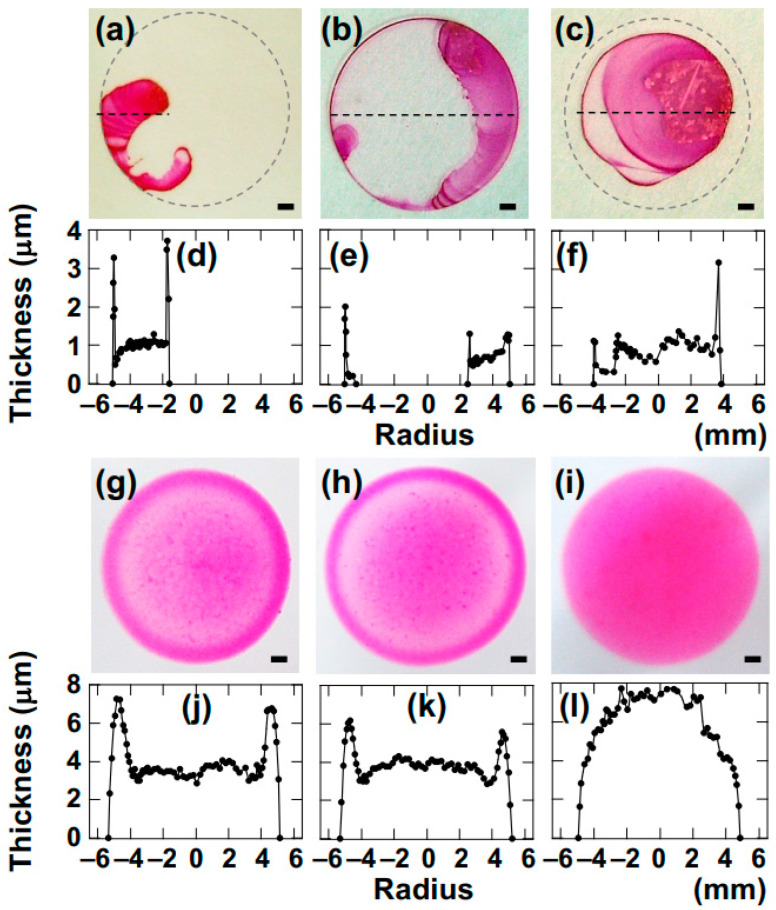
Drying patterns and cross-sectional profiles of R6G and R6G-mixed Sap aqueous dispersions at 25 °C. (**a**–**f**) *ϕ* = 0, [R6G] = 0.0001 mol/L. (**a**,**d**) [NaCl] = 0, (**b**,**e**) 0.001 mol/L, (**c**,**f**) 0.01 mol/L. (**g**–**l**) *ϕ* = 0.0029, [R6G] = 0.0001 mol/L. (**g**,**j**) [NaCl] = 0, (**h**,**k**) 0.001 mol/L, (**i**,**l**) 0.01 mol/L.

## Data Availability

Data are contained within the article.

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
