# Peer review of "Influence of Sol–Gel State in Smectite Aqueous Dispersions on Drying Patterns of Droplets"

_materials, 2024, doi:10.3390/ma17122891_

Round 1

Reviewer 1 Report

Comments and Suggestions for Authors

The author presents the influence of sol-gel state in smectite aqueous dispersions on drying patterns of droplets. Overall, the experiments were well conducted, and the characterizations were comprehensive and plausible. However, in the present form cannot been published. I think that this work has worthy for publication if following questions can be properly addressed.

1.     Sol-gel states are good explained, however; sol-gel in their complete form refer a method for synthesis of materials, 

In this case, What are the main chemical reaction involved in the sol and gel steps?. Please, represent these reactions. 

2.     Explain, why the quantities of the four types of synthetic smectite clay minerals were used on only one, are there references about these values? Justify this point.

3.     How many times were done the measurement of the dispersion? Please include the standard deviation of the results obtained. 

4.     Page 10 the summarizing result points must be included in conclusion section without repeat information. 

5.     Please, include recent references. 

6.     The English grammar must be revised. 

Comments on the Quality of English Language

The English grammar must be revised. 

Reviewer 2 Report

Comments and Suggestions for Authors

The authors of this paper investigated the drying state of four different clays forming dispersions and sol-gel states in water. In my opinion, the science in this article is weak, as only a few sets of simple experiments were conducted to analyze the phenomena without revealing the causes or considering the variability caused by different engineering environments on the solution drying patterns.

1. What are the specific engineering implications of the authors' study of drying patterns in clay water dispersions? Authors please give the engineering background in the introduction section.

2. What has the author done to build on previous research, please specify.

3. What are the concentration limits for the conversion of different clays from dispersed liquids to sol-gels?

4. Explain the reasons for the differences in the drying patterns of the dispersed phase and the sol-gel.

5. Please list the specific proportioning tables for the different solution sample designations in section 2.1, the textual representation appears confusing.

6. The conclusion section is too brief, giving only the experimental phenomenon without analyzing the cause of the phenomenon.

7. Give the situation of the solution-drying environment chosen for the article and explain the reasons for choosing that environment.

Comments on the Quality of English Language

Minor editing of English language required

Reviewer 3 Report

Comments and Suggestions for Authors

The submitted manuscript shows the influence of sol-gel state in smectite aqueous dispersions on the drying patterns of droplets. The Author has examined the influence of variables such as NaCl concentration and addition of two different pigments. The work is nicely written, presented and the results were properly discussed. I suggest the acceptance of this work after some minor revisions.

In the introduction, the definition of drying pattern should be introduced.

Adding a figure with some common deposition patterns to introduction would increase the clarity and improve the style of presentation.

In the introduction, it should be clearly explained why the suppression of the "coffee ring" phenomenon is so important.

Line 153-156, the units of those quantities should be stated.

Figure 5, the explanation of red tags should be included in the figure caption.

Line 304, it should be “The” not “he”

Line 299, the author writes “despite R6G being very small "molecules,"” – this should be rewritten, for example “despite the small size of R6G molecule”.

Line 299-300, this part should be evaluated as the vapor pressure depends not only on the size of the molecule but also on its ability to form intermolecular interactions and their strength

Reviewer 4 Report

Comments and Suggestions for Authors

The manuscript paper deals with the formation of drying patterns of different Smectite aquaeous dispersions. The authors have provided a sufficient introduction and explained the current state of knowledge in this field. The motivation for the study was the lack of research work in this area, with conflicting results from several authors. Adequate experimental techniques and method of processing the results were used. The presentation of the results is well handled. I think most of the paper is of a very good standard. 

I recommend that the authors make revisions based on the following comments. 

1- Provide a figure 

2- The conclusions part is not very conclusive. Revise this section to provide the following in addition to a recapitulation of the measured results: 

- clearly define the scientific contribution of your work

- Define the potential for application of your results. 

Reviewer 5 Report

Comments and Suggestions for Authors

See, please, the attached file

Comments on the Quality of English Language

English language should be a little bit performed

Round 2

Reviewer 2 Report

Comments and Suggestions for Authors

The summary and conclusion need to be further condensed.

Author Response

(Reviewer #2)

I am very grateful to reviewer 1 for the useful comments and suggestions that have helped me to improve my paper. As indicated in the responses that follow, based on these comments and suggestions, I carefully incorporated the necessary information into the revised version.

The conclusion has been made as concise as possible.

I once again deeply appreciate the cooperation of the Reviewer #2.

Sincerely yours,

Hiroshi KIMURA
